# Relationship between Perceived Physical Competence and Outdoor Play among Children Aged 9–12 Years-Focused Sex-Specific Differences

**DOI:** 10.3390/children10010135

**Published:** 2023-01-10

**Authors:** Ryo Goto, Kazufumi Kitagaki, Kana Horibe, Kazuya Tamura, Naoki Yamada, Rei Ono

**Affiliations:** 1Graduate School of Health Sciences, Kobe University, Kobe 654-0142, Japan; 2Faculty of Rehabilitation, Shijonawate Gakuen University, Osaka 574-0011, Japan; 3Department of Physical Activity Research, National Institutes of Biomedical Innovation, Health and Nutrition, Tokyo 162-8636, Japan

**Keywords:** physical self-perception, physical activity, sex characteristics

## Abstract

Background: Outdoor play (OP), which is considered important for children’s development, is declining every year. Perceived physical competence (PPC) is a vital factor that promotes physical activity such as OP, sports clubs, etc., but the relationship between PPC and OP was unknown. The purpose of this research was to investigate the relationship between PPC and OP in children and consider whether there were any sex-specific changes. Methods: A cross-sectional study was conducted in Japan with 288 children (134 girls, age: 10.6 ± 1.01 years). OP was assessed using an original self-report questionnaire. Each weekday, the children reported the time of OP and were classified as “high” if they played outside for at least an hour three times. PPC was evaluated with a self-report questionnaire developed by Okazawa et al. (1996). It has 12 questions and was assessed on a 5-point Likert scale. After adjusting for age, sex, BMI, screen time, sports club participation, and the number of friends, logistic regression analyses were carried out. Results: Children with better PPC were significantly more likely to be classified as “high” [crude odds ratio (OR): 1.04; 95% confidence interval (CI): 1.00–1.08; adjusted OR: 1.04; 95% CI: 1.00–1.08]. Only girls with better PPC were significantly more likely to be classified as “high” in a sex-based stratified analysis [crude OR: 1.08; 95% CI: 1.01–1.15, adjusted OR 1.09; 95% CI: 1.02–1.17]. Conclusions: Particularly among girls, OP could be promoted as a voluntary physical activity with improved PPC.

## 1. Introduction

Children’s daily physical activity is not only beneficial to their physical and mental health (body composition, physical fitness, quality of life or well-being, self-esteem, etc.) [1,2] but also about its carryover effect in adulthood [3,4]. However, approximately 80% of children and adolescents don’t meet the international physical activity guidelines about engaging in moderate-to-vigorous physical activity for 1 h/day [5] and recent year, a part of physical activity tend to decrease in children and adolescent [6]. Inactivity has become a global issue, with the need to increase physical activity beginning in childhood becoming essential.

Outdoor play (OP) is defined as “unstructured physical activity that takes place outdoors in the child’s free time” [7]. Daily physical activity in childhood includes physical education (PE) class, sports club participation, active transportation, and OP. OP accounts for the largest proportion of daily physical activity [8,9]. Aside from the health benefits of physical activity, there are numerous benefits associated with OP. OP provides children the opportunities to be creative, to learn how to structure games in the absence of specific rules, and to develop and change their experiences related to physical activity according to variable material and social contexts [10]. In other words, OP is believed to play an important role in children’s cognitive, physical, and emotional development [11]. However, there has been a decline in the frequency of OP among children since the 1970s [12]. Therefore, it is important to promote OP during childhood and adolescence.

Motor competence is defined as “a person’s ability to execute different motor acts, including coordination of fine and gross motor skills that are necessary to manage everyday tasks” [13]. Higher motor competence is known to be associated with higher physical activity [14]. On the other hand, perceived physical competence (PPC) is described as one’s perception of his or her actual motor competence level [15]. Like Stodden’s model, motor competence and PPC are both important in influencing the physical activity trajectories of children [16].

Various factors influence motivation for physical activity in children. There are personal factors (intrinsic motivation, extrinsic motivation, self-concept, and self-esteem), environmental factors (access to facilities, a safe environment), and social factors (peer support, parental support, and family support) [17,18,19]. Focusing on personal factors, PPC is considered a central determinant of physical activity behavior and is considered the strongest predictor of physical activity among the various aspects of self-concept [20,21]. A longitudinal study showed the long-term effect of PPC on physical activity from childhood into adolescence [22,23]. Adolescents with higher PPC were more motivated to participate in PE classes, were more physically active, and were more participative in sports even if the actual motor competence was low [24]. Therefore, PPC could be considered a vital factor in promoting physical activity since it occurs during childhood and adolescence.

A systematic review summarized the relationship between PPC and physical activity, which summarized that high PPC was positively associated with physical activity, and this relationship was subsequently investigated [25,26]. However, studying the relationship between PPC and physical activity does not verify whether children’s decisions are reflected in their physical activity. Furthermore, to the best of our knowledge, no study has investigated the association between PPC and OP among schoolchildren. The essential element of OP is discretionary choice [27], and PPC is considered the central determinant of behavior related to physical activity [25]. Therefore, it is possible to obtain a result that reflects children’s choices about physical activity by investigating the relationship between PPC and OP. Furthermore, the strength of the relationship between PPC and physical activity differs by sex [25], and OP is affected by different factors depending on sex [28,29,30]. Therefore, this study aimed to investigate the relationship between PPC and OP and to determine sex-specific differences.

## 2. Methods

This cross-sectional study recruited children in grades 4–6 from two public elementary schools in Kobe, Japan, in October 2020. The inclusion criteria for this study were individuals who agreed to take part in our study. The exclusion criteria were children with missing data. The authors explained the protocol of the study in writing and verbally to the children, the school principals, and in writing to the parents. Informed consent was obtained from both the children and their parents. This research protocol was approved by the Research Ethics Committee of Kobe University Graduate School of Health Science [approval number: 545-4].

We obtained data from self-reported questionnaires and anthropometric measurements. The children completed the questionnaire with the assistance of highly trained research staff. The elementary school teachers and school staff conducted anthropometric measurements during routine health checkups at the respective elementary school in June 2020.

We assessed OP based on self-reported questionnaire responses. To easily understand our questionnaire, we defined OP as “non-supervised physical activity outside with friends or family” [31]. The OP was assigned by asking the following questions: “What kind of outdoor play did you do after school on the last few weekdays?” and “And how many hours did you spend?” Each weekday, the children reported the content and time of OP and were classified as “high” if they performed OP for ≥1 h at least three times a week [31] and “low” if they performed OP less than three times a week.

PPC was evaluated with the self-report questionnaire prepared by Okazawa et al. [32]. This questionnaire comprises 12 questions for self-assessing positive recognition of athletic ability or attainment and the degree of subjective confidence, including environmental factors that support self-confidence related to physical exercise. For example, the scale includes the following questions: “I think I have excellent athletic ability”, “I am confident about exercise”, “I believe that with practice, my skills and records will surely improve”, and “I have a friend who invites me to exercise with him or her”. The Cronbach’s α value of this scale was 0.73–0.88 for the schoolchildren [32]. Each question is scored on a five-point scale of 1 (strongly disagree), 2 (disagree), 3 (undecided), 4 (agree), and 5 (strongly agree). A higher score indicates a better PPC. The total score of the 12 questions was used in the analysis.

Age, sex, screen time, sports club participation, and number of friends were assessed from the self-reported questionnaire. Children indicated daily screen time for each device, including TV/video, computer/video games, and mobile phone [33,34]. The total screen time was determined and categorized as ≤2 h/day or >2 h/day [35]. Information on sports club participation was collected using the following questions: “Do you belong to a sports club on weekdays?” The children answered the question with “yes” or “no”. As for the number of friends, we defined it as “those with whom you often spend recess, lunchtime, and after-school time”, and the children answered the number in an open-ended form. They were classified as “many” if they were above and "few" if they were below the median value. We obtained height (cm) and weight (kg) from the records of the anthropometric measurements at the elementary schools. Body mass index (BMI) was calculated by dividing weight in kilograms by height in meters squared (kg/m^2^).

Normality analysis was conducted using the Shapiro–Wilk test for continuous variables. Data for continuous variables are presented as the mean [standard deviation (SD)] or median [interquartile range (IQR)] and percentage for categorical variables. To assess the sex-specific difference in variables, the unpaired t-test or the Mann–Whitney U test were used for continuous variables, as appropriate, and the chi-square test was used for categorical variables. The association between PPC and OP was evaluated using univariate logistic regression analysis and multivariate logistic regression analysis. Further, odds ratios (OR) and 95% confidence intervals (CI) were calculated. The objective variable was OP, and the explanatory variable was PPC. We selected the following as relevant confounding variables: age, sex, BMI, screen time, sports club participation, and number of friends [27,36,37]. In stratified analysis, the same analysis was performed for each sex. To compare the results of the relationship between PPC and organized sports, we also evaluated the relationship between PPC and sports club participation using univariate logistic regression analysis and multivariate logistic regression analysis.

The level of significance (*p*) was set at <0.05. Data were analyzed using the free software R ver. 3.6.2 [38].

## 3. Results

We recruited 296 children and excluded 8 due to missing data. Finally, 288 children were included in the analysis (mean age ± SD: 10.60 ± 1.01 years; boys, *n* = 154, and girls, *n* = 134).

Table 1 shows the study sample characteristics for the total participants and each sex. Fifty children (17.4%) were categorized as “high” in OP, and the median [IQR] value for PPC was 45 [38, 51]. There was no significant difference between both sexes in terms of the frequency of OP and PPC.

The results of univariate and multivariate logistic regression analyses for the total number of participants are shown in Table 2. In a univariate logistic regression model, the children who were categorized as “high” in OP had significantly higher PPC (OR: 1.04; 95% CI: 1.00–1.08). After adjustment for age, sex, BMI, screen time, sports club participation, and number of friends, higher PPC remained significantly and independently associated with the frequency of OP (OR: 1.04; 95% CI: 1.00–1.08).

According to the results of the stratified analysis by sex (Table 3), girls who were categorized as “high” in OP had significantly higher PPC in both univariate (OR: 1.08; 95% CI: 1.01–1.15) and multivariate (OR: 1.09; 95% CI: 1.02–1.17) logistic regression analyses. However, there was no association between PPC and OP in boys (OR: 1.01; 95% CI: 0.97–1.06).

There was no significant association between PPC and sports club participation for all the participants in both the univariate (OR: 1.02; 95%CI: 1.00–1.05) and multivariate (OR: 1.01; 95%CI: 0.99–1.04) logistic regression analyses. There was also no significant relationship between PPC and sports club participation in the stratified analysis. In boys, the results were as follows: univariate logistic regression analysis (OR: 1.03; 95%CI: 0.99–1.06) and multivariate logistic regression analysis (OR: 1.01; 95%CI: 0.97–1.05). In girls, the results were as follows: univariate logistic regression analysis (OR: 1.01; 95%CI: 0.97–1.05) and multivariate logistic regression analysis (OR: 1.01; 95%CI: 0.97–1.05).

## 4. Discussion

In this study, we investigated the association between PPC and OP in children aged 9–12 years. We found that children who had higher PPC had a significantly high frequency of OP (>1 h at least three times a week), even when adjusted for age, sex, BMI, screen time, sports club participation, and number of friends. There was a significant association between PPC and OP, which differed by sex, i.e., only among girls but not among boys. This study was the first to clarify the relationship between PPC and physical activity, with a particular focus on OP. In other words, there was a significant association between PPC and OP, but there was no significant association between PPC and sports club participation in this study.

The results of this study are consistent with those of previous studies on the relationship between PPC and the physical activity domain [25]. Higher PPC may also improve PA through its association with other factors. In the personal factor domain, high PPC is an intrinsic motivation for physical activity [39]. In the social factor domain, PPC is positively related to parental and peer support [40]. Therefore, PPC may be directly and indirectly associated with physical activity.

As mentioned above, there is a significant association between PPC and PA; although there was a significant association between PPC and OP, there was no significant association between PPC and sports club participation in this study. PPC is the central determinant of behavior [25], and in the physical activity domain, it is generally interpreted as the confidence to play sports and outdoor games [20,21]. Competitive sports and highly structured activities, on the other hand, have been identified as barriers to children’s physical activity participation [17]. Therefore, in this study, there was only a significant association between PPC and OP, which is unstructured physical activity, but not sports club participation. And we believe that we obtained more reasonable results that reflect children’s psychological situation regarding physical activity.

This study revealed a significant association between PPC and OP only among girls. This result is consistent with a study that showed a direct path from PPC to MVPA only in girls using structural equation models [41]. This may explain the sex-specific differences in the frequency of OP noted in the present study. Girls had less outdoor playtime than boys, but girls had a higher proportion of outdoor playtime for all physical activities [9]. Therefore, OP was more representative of the physical activity-related characteristics of girls than of boys. The girls in this study had significantly lower sports club participation and screen time than boys (Table 1). Excessive participation in organized sports clubs has a direct negative impact on children’s free time [42]. Girls had more free time and less screen time, which may have made it easier for them to play outside in the limited time after school. However, our result was inconsistent with that of a previous study, wherein the association between PPC and physical activity was stronger in boys than in girls [25]. Furthermore, there is limited evidence of sex-specific differences in this relationship; thus, further research is warranted.

It is considered that PPC can be improved by interacting with others. Teachers’ feedback to children in PE classes, as well as interventions to improve their actual motor skills, have been shown to improve their PPC [43,44]. As both PPC and actual motor competence are lower in girls than boys during childhood and adolescence [14,45,46], there is room for intervention, especially in girls. Therefore, if PPC is improved in PE classes, there is a possibility that OP, which is voluntary physical activity, will increase. This effect may be greater for girls.

This study has some limitations. First, this study had a cross-sectional design, so we could only investigate the relationship between PPC and OP but not state the causal relationship. Moreover, there were reciprocal relationships among PPC and MVPA [31]. Therefore, future studies should investigate a longitudinal association between PPC and OP. Second, there are no standardized measurements of OP [47]. The children answered questions related to OP with the support of the study researchers. Therefore, recall bias was considered minimal. Third, this study was conducted during the COVID-19 epidemic. A decrease in PA in children has been reported during the COVID-19 epidemic [48]. On the other hand, during the lockdown, sports activities decreased while playing outside increased [49]. Although this study was conducted after the lockdown was lifted, it was possible that some sports groups were still affected by the lockdown; therefore, the association between PPC and PA and PPC and sports club participation may not have been accurately assessed. Finally, we only included two elementary schools and could not measure covariate factors such as actual motor competence, parental information, and environmental factors [50,51]. Future research needs to be comprehensive, including not only children’s information but also environmental information and types of OP and choice.

## 5. Conclusions

This study revealed the relationship between PPC and OP among children aged 9–12 years. In today’s world, where children’s physical activity level is insufficient, improving PPC may promote OP as part of children’s voluntary physical activity, especially among girls. School teachers should focus on PPC to promote the autonomous movement of children.

## Figures and Tables

**Table 1 children-10-00135-t001:** Characteristics of the study population.

	Total (*n* = 288)	Boys (*n* = 154)	Girls (*n* = 134)	*p*
Age (years)	10.60 ± 1.01	10.55 ± 1.07	10.67 ± 0.94	0.291
Height (cm)	142.18 ± 8.82	142.52 ± 9.00	141.78 ± 8.62	0.483
Weight (kg)	36.65 ± 9.53	37.21 ± 10.07	36.00 ± 8.86	0.283
BMI (kg/m^2^)	17.89 ± 3.08	18.06 ± 3.28	17.69 ± 2.83	0.307
Screen time (>2 h/day)	210 (72.9)	121 (78.6)	89 (66.4)	0.024
Sports club participation (yes)	119 (41.5)	74 (48.1)	45 (33.8)	0.016
Number of friends (many)	137 (47.6)	88 (57.1)	49 (36.6)	0.001
Perceived physical competence	45 (38, 51)	46 (39, 52)	44 (37, 50)	0.148
Outdoor play (high)	50 (17.4)	30 (19.5)	20 (14.9)	0.351

Note: Data are presented as mean ± SD, *n* (%), or median [IQR]. *p*-values show the results of sex-specific differences based on an unpaired *t*-test or Mann–Whitney U test for continuous variables and the chi-square test for categorical variables. BMI: body mass index; SD: standard deviation; IQR: interquartile range.

**Table 2 children-10-00135-t002:** Logistic regression analysis assessing the relationship between perceived physical competence and outdoor play for all participants.

	Univariate		Multivariate	
	OR	95% CI	OR	95% CI
Perceived physical competence	1.04 *	1.00–1.08	1.04 *	1.00–1.08
Age (years)	-	-	1.19	0.86–1.66
Sex (girl)	-	-	0.74	0.38–1.44
BMI (kg/m^2^)	-	-	0.86 *	0.76–0.98
Screen time (>2 h/day)	-	-	1.52	0.70–3.29
Sports club participation (yes)	-	-	0.69	0.35–1.35
Number of friends (many)	-	-	1.21	0.62–2.35

Note: *: *p* < 0.05; OR: odds ratio; CI: confidence interval; BMI: body mass index.

**Table 3 children-10-00135-t003:** Logistic regression analysis assessing the relationship between perceived physical competence and outdoor play for each sex.

	Univariate		Multivariate	
	OR	95% CI	OR	95% CI
Boys				
Perceived physical competence	1.01	0.97–1.06	1.00	0.96–1.05
Age (years)	-	-	1.11	0.74–1.65
BMI (kg/m^2^)	-	-	0.90	0.78–1.04
Screen time (>2 h/day)	-	-	1.05	0.37–2.98
Sports club participation (yes)	-	-	0.91	0.39–2.13
Number of friends (many)	-	-	1.62	0.67–3.95
Girls				
Perceived physical competence	1.08 *	1.01–1.15	1.09 **	1.02–1.17
Age (years)	-	-	1.48	0.82–2.69
BMI (kg/m^2^)	-	-	0.79 *	0.63–0.99
Screen time (>2 h/day)	-	-	2.39	0.71–7.99
Sports club participation (yes)	-	-	0.42	0.12–1.41
Number of friends (many)	-	-	0.81	0.28–2.36

Note: *: *p* < 0.05; **: *p* < 0.01; OR: odds ratio; CI: confidence interval; BMI: body mass index.

## Data Availability

Data are available upon request to the corresponding author.

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
