# Peer review of "Relationship between Perceived Physical Competence and Outdoor Play among Children Aged 9–12 Years-Focused Sex-Specific Differences"

_children, 2023, doi:10.3390/children10010135_

Round 1
Reviewer 1 Report
Title: Relationship between perceived physical competence and out-door play among children aged 9–12 years
Article Type: Article
Summary
In this cross-sectional study, the authors investigated the relationship between perceived physical competence (PPC) and outdoor play (OP) among both girls and boys. They also identified any sex-specific differences. Participants were 288 children (both gender) from Japan. perceived physical competence (PPC) and outdoor play (OP) were assessed by self-report questionnaire. The results indicated that children with better PPC had a significantly higher probability of being categorized as “High”. In stratified analysis by sex, only girls with better PPC had a significantly higher probability of being categorized as “High”.
Evaluation
The topic of this study is interesting. The sample size and the design for the study is appropriate to answer the research questions, and the paper is well written. However, there are some points should be addressed by the authors, in order to improve the quality of the manuscript.
Points
- Please add a background to the abstract, for example, a description about the variables.
- L 14, (134 girls, age: 10.6 ± 13 1.01 years)
- One purpose of the study was to identify any sex-specific differences, please add this also to the title of the paper.
- Please add more references to the introduction about the topic and studies variables, there are lots of study related to your study but you didn't mention it.
- Please add the exclusion criteria too.
- How did you select the participants? What was the sampling method? How you did calculate sample size? Please clarify this in the method section.
- L 84, I think you should reverse it, “for ≥1 h”.
- Is it possible that the COVID-19 epidemic has affected your findings? If possible, please mention it in the limitations section.
Author Response
Thank you very much for reviewing our manuscript and offering valuable advice.
We have addressed your comments point by point and revised the manuscript accordingly. The revised manuscript (with or without highlight for changes) is submitted as attached.
We look forward to hearing from you regarding our revised manuscript and would be glad to respond to any further questions and comments if you have.
Sincerely yours,
Ryo Goto
Point 1: Please add a background to the abstract, for example, a description about the variables.
Response 1: Thank you for your comment. The following sentences have been added to the abstract.
Abstract (Page1- Line12-15)
Outdoor play (OP), which is considered important role for children's development, is declining every year. Perceived physical competence (PPC) is a vital factor that promote physical activity such as OP, sports club and etc, but the relationship between PPC and OP was unknown.
Point 2: L 14, (134 girls, age: 10.6 ± 13 1.01 years)
Response 2: Thank you for your comment. I have modified as you indicated.
Point 3: One purpose of the study was to identify any sex-specific differences, please add this also to the title of the paper.
Response 3: Thank you for your comment. The following words have been added to the title.
Relationship between perceived physical competence and outdoor play among children aged 9–12 years -Focused sex-specific differences-
Point 4: Please add more references to the introduction about the topic and studies variables, there are lots of study related to your study but you didn't mention it.
Response 4: Thank you for your comment. I have added some references related to this study to the introduction.
Reference:
- Saunders TJ, Gray CE, Poitras VJ, et al. Combinations of physical activity, sedentary behaviour and sleep: relationships with health indicators in school-aged children and youth. Appl Physiol Nutr Metab. 2016; 41(6 Suppl 3): S283-93. doi: 10.1139/apnm-2015-0626.
- García-Hermoso A, Izquierdo M, Ramírez-Vélez R. Tracking of physical fitness levels from childhood and adolescence to adulthood: a systematic review and meta-analysis. Transl Pediatr. 2022; 11(4): 474-486. doi: 10.21037/tp-21-507.
- Booth VM, Rowlands AV, Dollman J. Physical activity temporal trends among children and adolescents. J Sci Med Sport. 2015; 18(4): 418-25. doi: 10.1016/j.jsams.2014.06.002.
Point 5: Please add the exclusion criteria too.
Response 5: Thank you for your comment. The following text has been added to the text.
Methods (Page2; Line87-88)
The exclusion criteria were children with missing data.
Point 6: How did you select the participants? What was the sampling method? How you did calculate sample size? Please clarify this in the method section.
Response 6: Thank you for your comment. The sample size was not calculated for this study, as it is the maximum number of children from two cooperating elementary schools, which is the maximum number possible in a feasible facility.
Point 7: L 84, I think you should reverse it,“for ≥1 h”.
Response 7: Thank you for your comment. This category means children ategorized as “High” who performed OP for ≥1h at least three times a week, and “Low” who less than three times a week. So it is correct “for ≥1h”, but the wording is difficult to understand, change the text as follows.
Methods(Page3; Line101-103)
The children described the content and time of OP on each weekday and were categorized as ”High” if they performed OP for ≥1 h at least three times a week [31] and ”Low” if they performed OP less than three times a week.
Point 8: Is it possible that the COVID-19 epidemic has affected your findings? If possible, please mention it in the limitations section.
Response 8: Thank you for your comment. As you pointed out, the impact of COVID-19 epidemic is considered, I have added the following sentences to the limitations.
Discusion(Page6; Line229-235)
Third, this study was conducted during covid-19 epidemic. A decrease in PA in children has been reported under the covid-19 epidemic [49]. On the other hand, during the lockdown, sports activties decreased while playing outside increased [50]. Although this study was conducted after the lockdown was lifted, some sports groups were still affected by the lockdown, and the association between PPC and PA(OP and organized sports) may not have been accurately assessed.
Reviewer 2 Report
Thank you for the opportunity to review this paper. There is great scope to explore the use of outdoor play in addressing low physical activity levels of children globally. The study small but is sound, and the paper is mostly well written and presented. Some major matters to address:
1. A deeper insight into the factors influencing engagement in and motivation for PA / OP in children and adolescents is needed in the introduction (e.g., opportunity, access, choice, variety, environmental factors, social factors, self-esteem / perceptions etc.). Then detail on how the findings of this study relate to these factors should be discussed later in the paper. If we want to promote OP for improving PPC then all the factors facilitating these should be presented. Further, the literature you draw from should focus on children - given your sample includes 10 year old children - not adolescents - they tend to have different factors influencing PA participation.
2. Given that the paper focuses in PPC and you have the data regarding organised sport, I am unsure as to why this data was not compared to OP. Is the association between PPC and OP and different to that of OP and Organise sport (and between girls / boys etc). This information would be useful.
I have included some specific comments throughout the manuscript in the uploaded document.

Author Response
Thank you very much for reviewing our manuscript and offering valuable advice.
We have addressed your comments point by point and revised the manuscript accordingly. The revised manuscript (with or without highlight for changes) is submitted as attached.
We look forward to hearing from you regarding our revised manuscript and would be glad to respond to any further questions and comments if you have.
Sincerely yours,
Ryo Goto
Point 1: A deeper insight into the factors influencing engagement in and motivation for PA / OP in children and adolescents is needed in the introduction (e.g., opportunity, access, choice, variety, environmental factors, social factors, self-esteem / perceptions etc.). Then detail on how the findings of this study relate to these factors should be discussed later in the paper. If we want to promote OP for improving PPC then all the factors facilitating these should be presented. Further, the literature you draw from should focus on children - given your sample includes 10 year old children - not adolescents - they tend to have different factors influencing PA participation.
Response 1: Thank you for your comment. I have added the sentences the factor of influencing engagement in motivation for PA to the introdcution and discussion.
Introducion (Page2; Line60-63)
Various factors influence motivation in physical activity in children. There are personal factors (intrinsic motivation, extrinsic motivation, self-concept, self-esteem), environmental factors (access to facilities, safe environment), social factors (peer support, parental support, family support) [17-19].
Discussion(Page5; Line184-189).
In other words, there was a significant association between PPC and OP, there was no significant association between PPC and sports club participation in this study.
And I have added some references to include 10years old children in the study design.
Point 2: Given that the paper focuses in PPC and you have the data regarding organised sport, I am unsure as to why this data was not compared to OP. Is the association between PPC and OP and different to that of OP and Organise sport (and between girls / boys etc). This information would be useful.
Response 2: Thank you for your comment. I have added the sentences for comparison of the results of the relationship between PPC and OP and the relationship between PPC and sports club participation. The added points are as follows
Methods (Page3; Line137-140)
For to compare the results of relationship between PPC and organized sports, we also evaluated between PPC and sports club participation using univariate logistic regression analysis, followed by multivariate logistic regression analysis.
Results(Page4; Line165-172)
There were no significant association between PPC and sports club participation for all the participant in both univariate (OR: 1.02; 95%CI: 1.00-1.05) and multivariate (OR: 1.01; 95%CI: 0.99-1.04) logistic regression analysis. Also in stratified analysis, there were no significant association between PPC and sports club participation. In boys, the results were follows: univariate logistic regression analysis (OR: 1.03; 95%CI: 0.99-1.06), and multivariate logistic regression analysis (OR: 1.01; 95%CI: 0.97-1.05). In girls, the results were follows: univariate logistic regression analysis (OR: 1.01; 95%CI: 0.97-1.05), and multivariate logistic regression analysis (OR: 1.01; 95%CI: 0.97-1.05).
Discussion(Page5; Line181-183.)
In other words, there was a significant association between PPC and OP, there was no significant association between PPC and sports club participation in this study.
Discussion(Page6; Line190-198))
As mentioned above, there is a significant association between PPC and PA, although there was a significant association between PPC and OP, there was no significant association between PPC and sports club participation in this study. PPC is the central determinant of behavior [25], and in the physical activity domain, it is generally operationalized as the confidence to perform sports and outdoor games [20,21]. On the other hand, competitive sports and highly structured activities were reported barriers to physical activity participation in children [17]. Therefore, in this study, there was only a significant association between PPC and OP which is unstructured physical activity, but not sports club participation.
And the table below are the results of logistic regression analysis for the relationship between PPC and sports club participation. Thank you for your confirmation.
|
Table 4. Logistic regression analysis assessing the relationship between perceived physical competence and sport club participatin for all participants |
|||||
|
|
Univariate |
|
Multivariate |
||
|
|
OR |
95% CI |
|
OR |
95% CI |
|
Perceived physical competence |
1.02 |
1.00–1.05 |
|
1.01 |
0.99–1.04 |
|
Age (years) |
- |
- |
|
1.13 |
0.89–1.45 |
|
Sex (girl) |
- |
- |
|
0.50** |
0.30–0.84 |
|
BMI (kg/m2) |
- |
- |
|
0.99 |
0.91–1.08 |
|
Screen time (>2 h/day) |
- |
- |
|
0.42** |
0.24–0.75 |
|
Outdoor play (High) |
- |
- |
|
0.70 |
0.36–1.35 |
|
Number of friends (Many) |
- |
- |
|
1.29 |
0.77–2.14 |
|
Note: **: p < 0.01; OR: odds ratio; CI: confidence interval; BMI: body mass index |
|||||
|
Table 5. Logistic regression analysis assessing the relationship between perceived physical competence and sports club participation for each sex |
|||||
|
|
Univariate |
|
Multivariate |
||
|
|
OR |
95% CI |
|
OR |
95% CI |
|
Boys |
|
|
|
|
|
|
Perceived physical competence |
1.03 |
0.99–1.06 |
|
1.01 |
0.97–1.05 |
|
Age (years) |
- |
- |
|
1.39* |
1.00–1.94 |
|
BMI (kg/m2) |
- |
- |
|
0.99 |
0.89–1.10 |
|
Screen time (>2 h/day) |
- |
- |
|
0.28** |
0.11–0.69 |
|
Outdoor Play (High) |
- |
- |
|
0.93 |
0.39–2.17 |
|
Number of friends (Many) |
- |
- |
|
1.37 |
0.68–2.78 |
|
Girls |
|
|
|
|
|
|
Perceived physical competence |
1.01 |
0.97–1.05 |
|
1.01 |
0.97–1.05 |
|
Age (years) |
- |
- |
|
0.84 |
0.56–1.25 |
|
BMI (kg/m2) |
- |
- |
|
0.99 |
0.86–1.14 |
|
Screen time (>2 h/day) |
- |
- |
|
0.59 |
0.27–1.31 |
|
Outdoor Play (High) |
- |
- |
|
0.41 |
0.12–1.37 |
|
Number of friends (Many) |
- |
- |
|
1.15 |
0.54–2.48 |
|
Note: *: p < 0.05; **: p < 0.01; OR: odds ratio; CI: confidence interval; BMI: body mass index |
|||||
Point 3: I have included some specific comments throughout the manuscript in the uploaded document..
Response 3: Thank you for your specific comments. For each comment, I have written a reply to the comment and a revision of the text in the uploaded file.